# Analysis of the effects of a tricyclic antidepressant on secondary sleep disturbance induced by chronic pain in a preclinical model

Hisakatsu Ito⊙*, Yoshinori Takemura‡, Yuta Aoki☉, Mizuki Hattori☉, Hideyo Horikawa‡, Mitsuaki Yamazaki☉

Department of Anesthesiology, University of Toyama, Toyama, Japan

☉ These authors contributed equally to this work.
‡ These authors also contributed equally to this work.
* hisa@med.u-toyama.ac.jp

**Data Availability Statement:** All relevant data are within the manuscript files.

**Funding:** HI received grant below. Grant numbers: 26893092 Grant name:Grant-in-Aid for Research

## Abstract

Chronic pain and sleep have a bidirectional relationship that promotes a vicious circle making chronic pain more difficult to treat. Therefore, pain and sleep should be treated simultaneously. In our previous study, we suggested that hyperactivation of ascending serotonergic neurons could cause secondary sleep disturbance in chronic pain. This study aimed to demonstrate the effects of a tricyclic antidepressant (amitriptyline) and a selective 5-hydroxy-tryptamine 2A (5-HT$_{2A}$) antagonist (MDL 100907) that adjust serotonergic transmission, on secondary sleep disturbance induced in a preclinical chronic pain model. We produced a chronic neuropathic pain model by partial sciatic nerve ligation in mice, analyzed their electroencephalogram (EEG) and electromyogram (EMG) using the SleepSign software, and evaluated the sleep condition of the pain model mice after administration of amitriptyline or MDL 100907. Amitriptyline improved thermal hyperalgesia and the amount of sleep, especially non-REM sleep. Time change of normalized power density of δ wave in the nerve ligation group with amitriptyline administration showed a normal pattern that was similar to sham mice. In addition, MDL 100907 normalized sleep condition similar to amitriptyline, without improvement in pain threshold. In conclusion, amitriptyline could improve sleep quantity and quality impaired by chronic pain. 5-HT$_{2A}$ receptor antagonism could partially contribute to this sleep improvement, but is not associated with pain relief.

## Introduction

Sleep disturbance in chronic pain patients is considered an important public health problem. 50–80% of chronic pain patients report some kind of sleep problem [1–4]. On the other hand, sleep loss causes increased pain intensity and spontaneous pain [5–7]. This bidirectional relationship between sleep and pain promotes a vicious circle that makes the pathology of chronic pain more intractable and complex. Furthermore, sleep loss impairs cognition, decision

Activity start-up from Japan Society for the Promotion of Science Website: https://kaken.nii.ac.jp/ja/ The funders had no role in study design, data collection and analysis, decision to publish, or preparation of the manuscript.

**Competing interests:** The authors have declared that no competing interests exist.

making, motor function, and mood [8, 9]. In addition, poor sleep cause obesity, diabetes, and hypertension, that are considered risk factors for stroke and fatal cardiovascular disease [10–12]. Therefore, the exploration of sleep treatment in chronic pain patients has clinical significance.

Commonly prescribed sedatives, such as benzodiazepines and non-benzodiazepines, induce sleep through the allosteric regulation of gamma-aminobutyric acid A ($GABA_A$) receptors. It has been reported that this pharmacological effect could be limited in cases of secondary sleep disturbance induced by pain [13]. Indiscriminate use of these drugs can cause a number of side effects, such as sedation, daytime sleepiness, dizziness, and cognitive and motor impairments [14, 15].

Amitriptyline, a tricyclic antidepressant, is a first-line drug for neuropathic pain, and is commonly prescribed for the treatment of neuropathic pain. A previous clinical study showed that amitriptyline, which adjusts serotoninergic transmission in the brain, could be effective in sleep disturbance due to chronic neuropathic pain [16]. However, it is not known whether this sleep-improving effect is an indirect effect due to pain improvement or a direct pharmacological effect of amitriptyline.

Amitriptyline has several mechanisms of action, such as serotonin receptor antagonism, adrenalin $\alpha_1$ receptor antagonism, histamine-1 receptor antagonism, muscarine-1 receptor antagonism, and sodium channel blocker [17–20]. There is a possibility that these mechanisms directly improve sleep, especially of 5-hydroxytryptamine 2A ($5\text{-HT}_{2A}$) receptor antagonism by amitriptyline. [21, 22]. The reticular activating system in the brain stem is a crucial neuronal network that controls sleep and wakefulness. Five neurotransmitters, serotonin, dopamine, noradrenaline, histamine, and acetylcholine are considered to be associated with wakefulness. We suggested previously that hyperactivation of ascending serotonergic neurons could cause secondary sleep disturbance in chronic pain [23]. Therefore, we hypothesized that antagonism of excitatory $5\text{-HT}_{2A}$ receptors by amitriptyline would improve pain-related sleep disturbance directly. In the present study, we investigated the effects of amitriptyline on secondary sleep disturbance induced in a preclinical model of chronic neuropathic pain, by analyzing the electroencephalogram (EEG) and electromyogram (EMG). Furthermore, we investigated the effects of $5\text{-HT}_{2A}$ receptor antagonism, which is one of the pharmacological effects of amitriptyline, on sleep condition in the same pain model.

## Materials and methods

### Animals

This study was carried out in strict accordance with the Act on Welfare and Management of Animals and the recommendations in the Guidelines for Proper Conduct of Animal Experiments of the Science Council of Japan. The protocol was approved by the Committee of Ethics in Animal Experiments of the University of Toyama (Protocol Number: A2014NED-14). All surgeries were performed under Isoflurane anesthesia, and all efforts were made to minimize suffering. Male C57BL/6J mice (8 weeks; Sankyo Labo Service, Japan) were used in this study. The animals were housed in a room maintained at constant temperature and humidity (24 ±2˚C, 50±10%), with a 12 h light -dark cycle (7:00, 19:00). Food and water were available *ad libitum*. Every effort was made to minimize the numbers and any suffering of animals used in the experiments. Each animal was used only once.

### Chronic neuropathic pain mice model

We produced a chronic neuropathic pain model by partial sciatic nerve ligation under general anesthesia with 3% isoflurane as described previously by Seltzer et al. [24]. Only half of the

thickness of the common sciatic nerve on the right-side hind thigh of C57BL/6J male mice were ligated by 8–0 surgical silk. In sham operated mice, the sciatic nerve was only exposed without ligation.

### Evaluation of hyperalgesia

We performed plantar test to evaluate the thermal hyperalgesia as reported in our previous study [23]. The tests were performed following vehicle administration on the $7^{th}$ post-operative day, and following administration of amitriptyline or the selective 5-HT$_{2A}$ receptor antagonist, MDL 100907, on the $8^{th}$ post-operative day. We placed mice in an acrylic cylinder (height: 15 cm, Diameter: 8 cm) following vehicle or drug administration. Mice were habituated for 1 hour before measurement. The mice's hind paws were stimulated by infrared heat device (IITC Life Science Inc., USA) and the paw withdrawal latency was measured three times and the mean value was calculated. The paw withdrawal latency was defined as the time from thermal stimulation until the hind paw was withdrawn. The minimum interval between consecutive tests was 3 minutes. Paw movements with weight shift were excluded. The intensity of infrared radiation was adjusted so that the paw withdrawal latency was about 8–10 seconds in normal mice. We tested only the ipsilateral paw in the plantar tests to confirm the effects of sciatic nerve ligation surgery and drug administrations on pain in the injured paw.

### EEG and EMG recording

Under general anesthesia with 3% isoflurane, we implanted the electrodes and devices to record EEG and EMG and fixed them on the mice's skulls using a stereotaxic apparatus. Four EEG electrodes were placed on the dura mater through holes drilled in the skull, 1.5 mm lateral to the sagittal suture on either side of, and 1 mm anterior to coronal and lambda sutures. Two wire electrodes to record EMG were inserted under the trapezius muscle. They were fixed using a head mount device (Pinnacle Technology, USA) with acrylic lysine. Mice were administrated amitriptyline, MDL 100907, or vehicle, at 6:40 am and 18:40. EEG and EMG of the mice were recorded for 24 hours (7:00 am to next 7:00 am) through preamplifier and cable with a low-torque commutator, which allowed the mice unencumbered freedom of movement.

### Analysis of sleep condition

We analyzed the EEG and EMG using the SleepSign software (Kissei Comtec, Japan) and evaluated the sleep condition of the mice. 5 seconds were defined as 1 epoch and classified into awake, REM sleep, or non-REM sleep stages. Sleep stage was decided according to low and stable integral value of EMG amplitude. Further, the sleep stage was classified into REM sleep or non-REM sleep stage according to the ratio of θ waves (5.0–10.0 Hz) or power density of δ waves (0.65–4.5 Hz). In brief, non—REM sleep (low EMG and high EEG amplitude, high δ wave activity), and REM sleep (low EMG and low EEG amplitude, high rate of θ wave over 80%) were determined for each 5 s epoch, as described in our previous study [23]. Non-REM sleep time, REM sleep time, and awake time per 24 hours for each animal were calculated by SleepSign. The values were summed for each group and divided by the number of animals to calculate the mean for each group. Finally, we calculated the mean time per hour. Furthermore, we evaluated the time change of power density of δ wave to estimate the quality of sleep. Normalized power density of δ wave was calculated as a value every 3 hours for the total power density of δ wave during non-REM sleep per day.

### Drugs

A tricyclic antidepressant, amitriptyline (Sigma-Aldrich, USA) was dissolved in 0.9% saline just before injection. The dose of amitriptyline was decided as 10 mg/kg as suggested in previous reports [25, 26]. Mice received intraperitoneal injection of amitriptyline or vehicle (0.9% saline). Selective 5-HT$_{2A}$ receptor antagonist, MDL 100907 (Wako Pure Chemical Industries, Japan) was dissolved in ethanol and diluted with 0.9% saline to 3% ethanol concentration. The dose of MDL 100907 was decided as 3 mg/kg as suggested in a previous report [27]. Mice received intraperitoneal injection of MDL 100907 or vehicle (3% ethanol).

### Statistical analysis

All data were represented as the mean with standard error of the mean (SEM) or the mean only. The statistically significant differences between the groups were assessed with one-way ANOVA, or two-way ANOVA followed by the Bonferroni multiple comparisons test. All statistical analyses were performed with Prism software, version 6.0 b (GraphPad Software, USA).

## Results

### Thermal hyperalgesia in neuropathic pain model

We evaluated the thermal hyperalgesia in neuropathic pain model and sham operated group at 7, 8 and 9 days after nerve ligation surgery, using the plantar test. The paw withdrawal latency was decreased in the nerve ligation group at any time points compared with the sham group (Fig 1). From this result, we confirmed that sufficient pain was persistent in this model during the experimental period.

### Changes in hyperalgesia by amitriptyline

We produced the neuropathic pain model mice as described above on test day 0. Plantar tests were performed before nerve ligation surgery on test day 0, after vehicle administration on test day 7, and after administration of amitriptyline on test day 8 (Fig 2A). In the results, amitriptyline significantly improved the reduction of paw withdrawal latency by sciatic nerve ligation (Fig 2B).

### Effects of amitriptyline on sleep condition

Mice were implanted EEG/EMG recording head-mount and electrodes at 5 days after nerve ligation or sham surgery, and habituated in a special cage for at least 24 hours before recording.

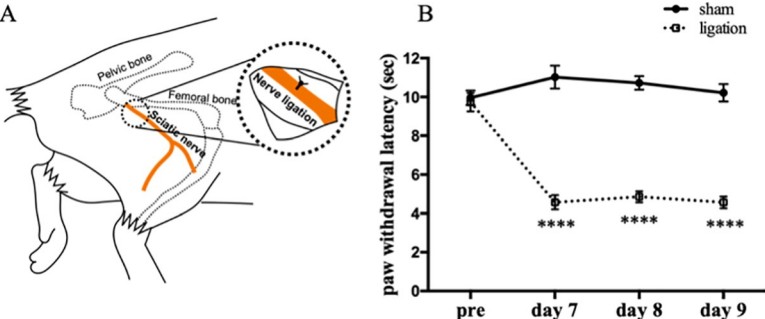

**Fig 1. Change of thermal hyperalgesia in neuropathic pain model mice.** Schematic diagram of sciatic nerve ligation (A). Plantar tests were performed before nerve ligation surgery, 7, 8 and 9 days after surgery. Data are expressed as means ± SEM. Data are analyzed using two-way ANOVA followed by Bonferroni's post hoc test and considered statistically significant at $^{****}P < 0.0001$ when comparing the sham (n = 8) and nerve ligation groups (n = 8) (B).

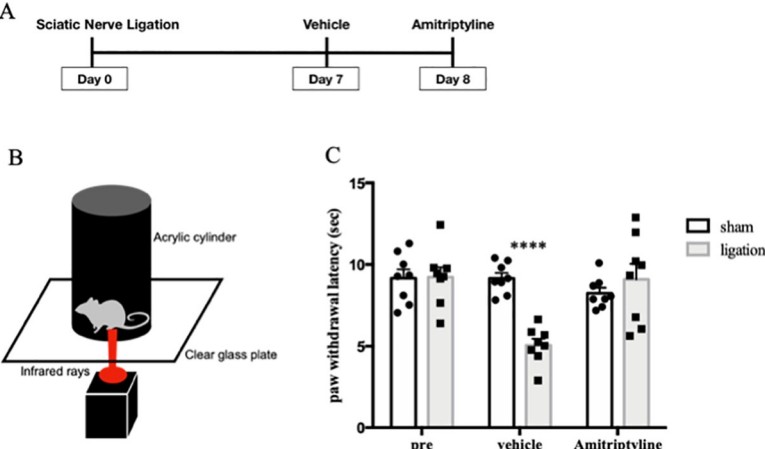

**Fig 2. Amitriptyline improved thermal hyperalgesia in neuropathic pain model.** Plantar tests were performed before nerve ligation surgery, test day 7 with vehicle, and test day 8 with amitriptyline administration (n = 8) (A). Schematic diagram of plantar test (B). Paw withdrawal latencies were plotted for every test (C). Data are expressed as means ± SEM. Data are analyzed using two-way ANOVA followed by Bonferroni's post hoc test and considered statistically significant at ****$P < 0.0001$ when comparing the sham and ligation groups.

We injected vehicle (0.9% saline, i.p.) at test day 7, and amitriptyline (10 mg/kg, i.p.) at test day 9, twice a day (at 7:00 and 19:00) (Fig 3A). In the results, the nerve ligation group showed significant increase in wake time and decrease in non-REM sleep time compared with sham group, when they received vehicle. Amitriptyline significantly reduced the wake time (2308 s/h vs. 1789 s/h; Ligation-vehicle vs. Ligation-amitriptyline, $p < 0.001$) and increased in non-REM sleep time (1175 s/h vs. 1654 s/h; Ligation-vehicle vs. Ligation-amitriptyline, $p < 0.001$) in the nerve ligation group. However, amitriptyline did not change the amount of sleep and wake time in the sham group (Fig 3B).

We analyzed the full frequency of EEG power density in non-REM and REM sleep. There were slight differences in peak of EEG power density during non-REM sleep when comparing

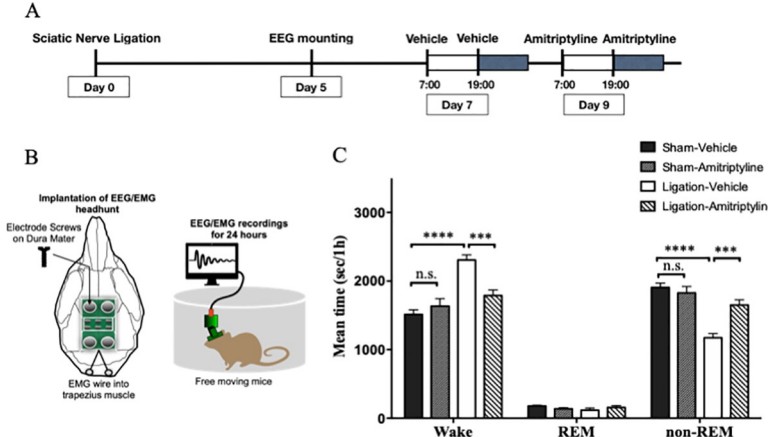

**Fig 3. Amitriptyline improved quantity of sleep in neuropathic pain model.** EEG and EMG were recorded for 24 hours with administration of vehicle (test day 7) and amitriptyline (test day 9) (A). Schematic diagram of EEG/EMG implantation (B). The mean value per hour of wake, REM sleep, and non-REM sleep time are represented for the sham (n = 7) and nerve ligation groups (n = 7) (C). Data are expressed as means ± SEM. Data are analyzed using two-way ANOVA followed by Bonferroni's post hoc test and considered statistically significant at ***$P < 0.001$, and ****$P < 0.0001$ when comparing the groups.

the sham-vehicle and ligation-vehicle (Fig 4A). We analyzed the time change of normalized power density of δ wave during non-REM sleep to evaluate the quality of sleep. Sham-vehicle mice showed high δ wave power, that indicated deep sleep at the beginning of the light phase. Then, their sleep became lighter towards the end of the light phase. On the other hand, sleep of nerve ligation mice was shallow in the beginning and deepened gradually until 16:00. However, there was no significant difference in the dark phase. With administration of amitriptyline, nerve ligation mice got deep sleep at first, in the inactive phase. Although there were no significant differences, amitriptyline tended to reduce the δ wave power in the active phase (Fig 4B).

**Changes in hyperalgesia by 5-HT$_{2A}$ receptor antagonist.**　Plantar tests were performed before sciatic nerve ligation surgery on test day 0, after vehicle administration on test day 7, and after administration of MDL 100907 on test day 8 (Fig 5A). In the results, MDL 100907 did not change the paw withdrawal latency in the neuropathic pain model mice (Fig 5B).

**Effects of 5-HT$_{2A}$ receptor antagonist on sleep condition.**　We injected vehicle on test day 7, and MDL 100907 (3 mg/kg, i.p.) on test day 9, twice a day (at 7:00 and 19:00) (Fig 6A). The nerve ligation group showed a significant increase in wake time and decrease in non-REM sleep time compared with the sham group, similar to the amitriptyline experiment. MDL 100907 significantly reduced the wake time (2171 s/h vs. 1526 s/h; Ligation-vehicle vs. Ligation-MDL, $p < 0.001$) and improved non-REM sleep time (1347 s/h vs. 1910 s/h; Ligation-

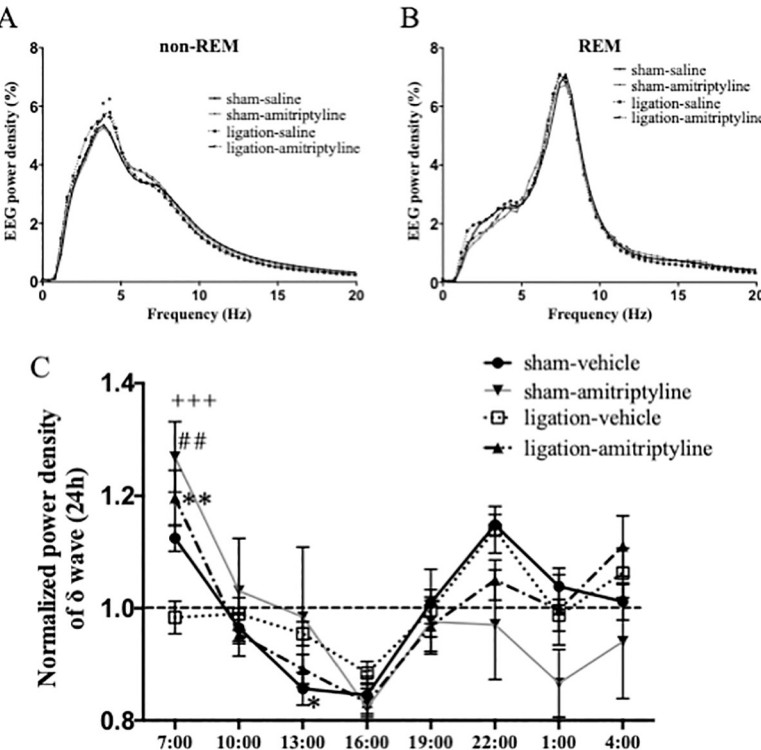

**Fig 4. Amitriptyline improved quality of sleep in neuropathic pain model.** Distribution of EEG power density in each frequency during non-REM and REM sleep. Data are expressed as means only. $^*P < 0.05$ at 4.2 Hz and 4.7 Hz when comparing sham-vehicle and ligation-vehicle groups (A). Time changes of the normalized power density of δ wave were plotted every 3 hours in sham (n = 5–7) and nerve ligation (n = 7) mice. Data are expressed as means ± SEM (B) and are analyzed using two-way ANOVA followed by Bonferroni's post hoc test and considered statistically significant at $^{**}P < 0.01$ when comparing sham-vehicle and ligation-vehicle groups, $^{##}P < 0.01$ when comparing ligation-vehicle and ligation amitriptyline group, and $^{+++}P < 0.001$ when comparing the sham-amitriptyline and ligation-vehicle groups.

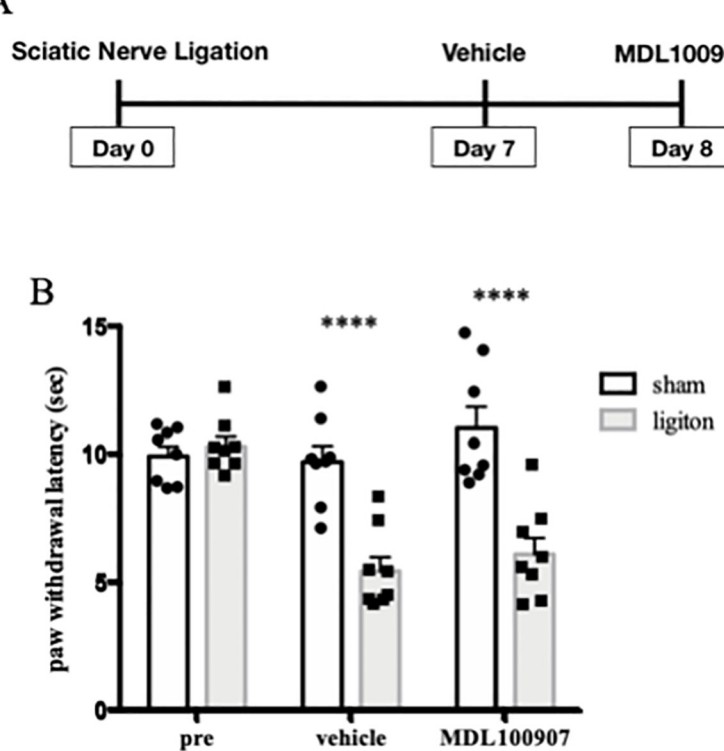

**Fig 5. Selective 5HT$_{2A}$ receptor antagonist improved thermal hyperalgesia in neuropathic pain model.** Plantar tests were performed before nerve ligation surgery, on test day 7 with vehicle, and on test day 8 with MDL 100907 administration (n = 8) (A). Paw withdrawal latencies were plotted for every test. Data are expressed as means ± SEM. Data are analyzed using two-way ANOVA followed by Bonferroni's post hoc test and considered statistically significant at $****P < 0.0001$ when comparing the sham and ligation groups (B).

vehicle vs. Ligation-MDL, $p < 0.001$) in the nerve ligation group (Fig 6B). In addition, MDL 100907 significantly increased REM sleep time in both sham and nerve ligation groups (Fig 6B).

The peak of EEG power density during non-REM sleep tended to be high in ligation-vehicle, but not significantly (Fig 7A). The result of normalized power density of δ wave during non-REM sleep showed that administration of MDL 100907 improved the sleep quality of nerve ligation mice, to obtain deep sleep at first and go to light sleep gradually, similar to amitriptyline (Fig 7B).

## Discussion

In the present study, the analysis of EEG and EMG showed that the neuropathic pain model mice exhibited increased wake time and decreased non-REM sleep in the chronic pain state. A tricyclic antidepressant, amitriptyline improved both increased arousal time and reduced non-REM sleep that were secondary to the pain, but did not show a significant change in the sham group. Hyperactivity of monoamine neurons is one of the mechanisms for secondary sleep disturbance [23, 28]. Since amitriptyline inhibits the re-uptake of serotonin and norepinephrine, it increases the concentrations of these neurotransmitters in the synaptic cleft. However, amitriptyline antagonizes excitatory 5-HT$_{2A}$ receptors coupled with Gq protein that induce spontaneous excitatory synaptic transmission. On the other hand, amitriptyline has little effect on the inhibitory 5-HT$_{1A}$ receptors coupled with Gi protein [21, 22]. Therefore, we believe that

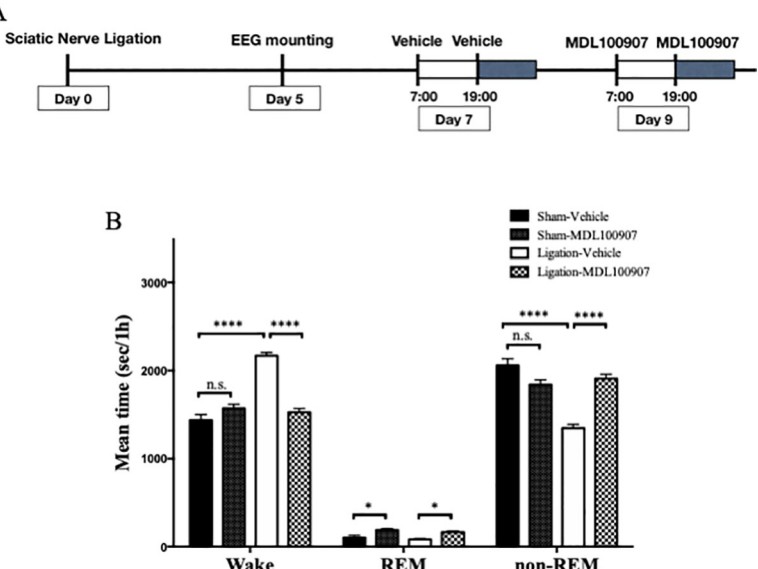

**Fig 6. Selective 5HT$_{2A}$ receptor antagonist improved quantity of sleep in neuropathic pain model.** EEG and EMG were recorded for 24 hours with administration of vehicle (test day 7) and MDL 100907 (test day 9) (A). The mean value per hour of wake, REM sleep, and non-REM sleep time are represented for sham (n = 7) and nerve ligation groups (n = 7). Data are expressed as means ± SEM. Data are analyzed using two-way ANOVA followed by Bonferroni's post hoc test and considered statistically significant at $^{***}P < 0.001$, and $^{****}P < 0.0001$ when comparing all groups (B).

amitriptyline may suppress the neuronal activity as an overall effect. Amitriptyline and selective 5-HT$_{2A}$ receptor antagonist MDL 100907 inhibit the binding of serotonin to 5-HT$_{2A}$ receptors [29]. This could result in a relative increase in serotonin binding to the inhibitory 5-HT$_{1A}$ receptor. These additive effects may suppress the hyperactivity of serotonergic neurons and obtain sleep normalization. This may be one of the reasons why amitriptyline does not cause over-sedation in normal subjects without pain.

According to the time change of normalized power density of δ wave during non-REM sleep, in the nerve ligation group with vehicle administration, the non-REM sleep is not deep enough in the beginning of the inactive phase. On the other hand, the nerve ligation group with amitriptyline administration showed a time change of normalized power density of δ wave similar to the sham-vehicle group in the inactive phase. These findings suggest that amitriptyline could provide near-normal quality of sleep in chronic pain. Amitriptyline tended to decrease the delta power at 22:00, though not significantly, in both the sham and ligation groups. Therefore, amitriptyline may induce shallow sleep in the active phase. However, since there is less sleep in the active phase, the data dispersion is large, and further research on this aspect is needed.

We performed additional experiments using the selective 5-HT$_{2A}$ receptor antagonist, Administration of 3 mg/kg MDL 100907 did not show inhibitory effects on hyperalgesia in the neuropathic pain model. However, the same dose of MDL 100907 showed a sleep-improving effect in the neuropathic pain model, similar to amitriptyline. Therefore, 5-HT$_{2A}$ antagonism was assumed to improve sleep directly, regardless of pain inhibition.

MDL 100907 slightly increased the time of REM sleep in both the ligation and sham groups in this study. There are few reports examining the effects of 5-HT$_{2A}$ receptor antagonists on REM sleep, and their results are controversial. Jaime M. et al reported that administration of 5-HT$_{2A}$ antagonist reversed the reduction of REM sleep caused by microinjection of 5-HT$_{2A/}$

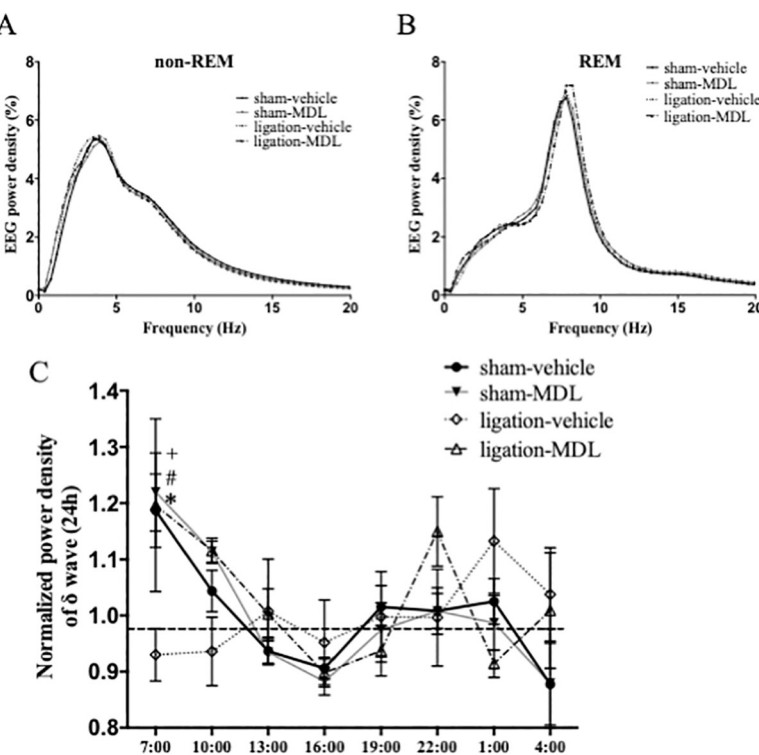

**Fig 7. Selective 5HT$_{2A}$ receptor antagonist improved quality of sleep in neuropathic pain model.** Distribution of EEG power density in each frequency during non-REM and REM sleep. Data are expressed as means only (A). Time changes of the normalized power density of δ wave are plotted every 3 hours in sham (n = 6) and nerve ligation (n = 5) mice. Data are expressed as means ± SEM (B). Data are analyzed using two-way ANOVA followed by Bonferroni's post hoc test and considered statistically significant at *$P < 0.05$ when comparing the sham-vehicle and ligation-vehicle groups, #$P < 0.05$ when comparing the ligation-vehicle and ligation-MDL 100907 groups, and +$P < 0.05$ when comparing the sham-amitriptyline and ligation-vehicle groups.

$_{2C}$ receptor agonist into the dorsal raphe nucleus in rats [30]. However, Stephen R. et al reported that 5-HT$_{2A}$ antagonist tended to increase REM sleep compared with vehicle administration, but not significantly [27].

There were slight differences in peak of EEG power density during non-REM sleep when comparing sham-vehicle and ligation-vehicle in the amitriptyline experiment (Fig 4A), but only a tendency in the MDL-experiment (Fig 7A). The reduction in sleep time with nerve injury may increase the absolute value of δ power density slightly, as a homeostatic compensation [31]. However, the results were not reproducible in our study.

This study has several limitations, First, as we did not find a significant abnormal behavior induced by the administration of these drugs, we cannot evaluate the neurological findings of the test in detail. Consequently, the possibility of neurological side effects such as suppression of motor coordination and activity cannot be ruled out. Second, the side effects of repeated administration were also not verified. Third, it is unknown whether these drugs had any effect on the ascending serotonergic neurons in brain stem reticular formation, because we only tested the effect of systemic administration. Fourth, only male mice were used for this study. The mechanisms of chronic pain have been shown to differ between males and females [32]; therefore, we need to examine them separately in pain studies. We consider that similar studies on female mice are also important as another study. Finally, we set the recovery time from EEG mounting surgery as 2 days, which was the same as in previous studies [13, 23]. However,

we could not prove the presence or absence of any effects of stress caused by the surgery and anesthesia.

Despite these limitations, our study has important implications for sleep improvement and pain relief. Further studies using advanced genetic engineering such as optogenetics or chemogenetics are required, to verify the neuronal mechanism in detail.

## Conclusions

This study showed that amitriptyline could improve sleep quantity and quality impaired by chronic pain. However, 5-HT$_{2A}$ receptor antagonism could not be associated with pain but was shown to be closely related to sleep improvement.

## Acknowledgments

We would like to thank Editage (www.editage.com) for English language editing.

## Author Contributions

**Conceptualization:** Yuta Aoki, Mitsuaki Yamazaki.

**Funding acquisition:** Hisakatsu Ito.

**Investigation:** Hisakatsu Ito, Mizuki Hattori, Hideyo Horikawa.

**Software:** Hisakatsu Ito.

**Supervision:** Yuta Aoki, Mitsuaki Yamazaki.

**Validation:** Mizuki Hattori.

**Writing – original draft:** Hisakatsu Ito.

**Writing – review & editing:** Yoshinori Takemura, Mizuki Hattori, Hideyo Horikawa, Mitsuaki Yamazaki.

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
