## [Decision Letter · Decision Letter 0]

11 Sep 2020

PONE-D-20-20832

Analysis of the effects of tricyclic antidepressant on secondary sleep disturbance induced by chronic pain

PLOS ONE

Dear Dr. Ito,

Thank you for submitting your manuscript to PLOS ONE. After careful consideration, we feel that it has merit but does not fully meet PLOS ONE’s publication criteria as it currently stands. Therefore, we invite you to submit a revised version of the manuscript that addresses the points raised during the review process.

We look forward to receiving your revised manuscript.

Kind regards,

Jianhong Zhou

Associate Editor

PLOS ONE

Journal Requirements:

2. Thank you for including your ethics statement:  "The present study was conducted in accordance with the Guiding Principles for the Care and Use of Laboratory Animals at University of Toyama (Toyama, Japan), as adopted by the Committee on Animal Research of University of Toyama.

Approval number: A2014NED-14

Methods of anesthesia: general anesthesia using isoflurane

Methods for euthanasia of animals: Carbon dioxide and Nerve destruction".   

Please amend your current ethics statement to confirm that your named ethics committee specifically approved this study.

For additional information about PLOS ONE submissions requirements for ethics oversight of animal work, please refer to http://journals.plos.org/plosone/s/submission-guidelines#loc-animal-research  

Reviewers' comments:

Reviewer's Responses to Questions

**Comments to the Author**

1. Is the manuscript technically sound, and do the data support the conclusions?

Reviewer #1: Yes

Reviewer #2: Partly

Reviewer #3: Yes

2. Has the statistical analysis been performed appropriately and rigorously? 

Reviewer #1: Yes

Reviewer #2: No

Reviewer #3: Yes

3. Have the authors made all data underlying the findings in their manuscript fully available?

Reviewer #1: Yes

Reviewer #2: No

Reviewer #3: No

4. Is the manuscript presented in an intelligible fashion and written in standard English?

Reviewer #1: No

Reviewer #2: No

Reviewer #3: Yes

5. Review Comments to the Author

Reviewer #1: The authors have provided an interesting study that contributes to the growing field of knowledge surrounding the relationship between disrupted sleep and chronic pain. Specifically, the authors sought to examine the role of serotonergic components of the ascending reticular activating system in sleep disruption induced by chronic pain. To accomplish this, a partial sciatic nerve ligation model was selected and a tricyclic antidepressant (amitriptyline) and a selective 5-HT2A antagonist (MDL 100907) were deployed. Using the plantar test and EMG/EEG recording, the authors observed that amitriptyline improved both hyperalgesia symptoms and sleep quantity, whereas MDL100907 only normalized sleep quantity but had no effects on examined hyperalgesia symptoms. From this study, we can conclude that that amitriptyline’s effects on improved sleep quality in rodents during chronic pain is likely due to at least some antagonism of 5-HT2AR, but that its ability to ameliorate hyperalgesia is mediated by another mechanism unrelated to 5-HT2AR antagonism. The authors provide a strong list of limitations to the interpretation of this study.

However, the authors need to seek out a native English speaker to improve the quality of the writing and grammar, need to explain why several groups appear to have been excluded from analysis (See Results comments), why proper controls were not used throughout (See Results comments), and why only males were used for the study (and please do not say because of the cyclic female sex steroid hormones unless they address the males’ pulsatile testosterone secretion patterns).

Comments:

Introduction:

- In the introduction, the authors should provide rationale for why they selected amitriptyline as the TCA of choice from the cited study (16). It may also be useful for the reader to know that it is commonly prescribed in the treatment of neuropathic pain.

- In the introduction, the rationale for using MDR 100907 should be explained. The third paragraph of the discussion may function best if moved to the introduction.

- Authors should provide justification for using only males in the study.

Methods:

- Did the mice receive any analgesic after surgery? If so, then the authors should describe this.

- The methods describing the evaluation of hyperalgesia are not consistent with the timelines in figures 2 & 3. Specifically, “Immediately after the administration of amitriptyline, selective 5-HT2A receptor antagonist, MDL100907, or vehicle, mice were placed in the acrylic cylinder (height: 15cm, Diameter: 8cm) and habituated for 1 hour before measurement.” These discrepancies should be corrected.

- What was the interstimulus interval for the repeated measurements of paw withdrawal latency? (E.g., 3-minutes were given between each test).

- The authors do not discuss ipsilateral or contralateral paw tests. Considering this information is not provided, we must assume that only the ipsilateral paw was tested. This should be clarified by the authors and a justification for not testing the contralateral paw should be provided.

- Though commonly used, the authors should cite the methods of classifying REM vs. NREM via Theta/Delta ratios, (e.g. Grosmark et. Al 2012 Cell).

Results:

- Figure 2 and figure 5 do not contain proper controls (i.e. a sham PSL group). The authors should provide a strong justification as to why this group was not included and a within-group historic control was used instead. Additionally, the authors should provide justification as to why a saline injection was given to the same group for a within-group historical comparison as opposed to having a full-factorial study design.

- Authors should delete word “clearly” from results section 1.

- Authors should provide rationale for why the sham-drug groups were excluded from figures 4 and 7.

- Authors should explain that delta power was increased during the inactive but not active phase. Reword this sentence “By administration of amitriptyline, nerve ligation mice got deep sleep at first and went to light sleep as same as sham mice” in Results section 3.

Discussion:

- Authors should reword the following sentence “According to the time change of normalized power density of δ wave during non-REM sleep, nerve ligation group with vehicle administration did not get enough had less deep sleep at an early sleep phase during the beginning of the inactive phase”

Reviewer #2: Antidepressants are reported to have analgesic effects, especially on chronic pain, and tricyclic antidepressants are used to treat chronic pain in combination with serotonin-noradrenaline reuptake inhibitors. Here, Ito and colleagues used the amitriptyline, a tricyclic antidepressant, and MDL 100907, a selective 5-HT2A antagonist, affect the reduction of chronic pain-induced sleep disturbance.

My major concern regarding this manuscript is about the experimental design. The authors confirmed the paw withdrawal latency of thermal hyperalgesia in neuropathic pain model mice in a weekly manner (7 days apart). However, in the subsequent experiments, they recorded the effect either on the next day (day 7, 8) or after of gap of 2 days (day7, 9) of the vehicle administration. Authors can follow any of the following three alternatives: Providing evidence that the mouse showed the same paw withdrawal latency within the days; or re-perform the experiment as a crossover manner, and/or collect the data weekly. Moreover, the authors have given less than 2 days for recovery after EEG mounting which might not be enough. The data presented to show the effect of the drug on sleep is also not convincing. Please provide a full frequency of EEG analysis in each vigilance stage of vehicle administration. Please provide the details on how they calculate the mean time used in each figure.

Minor Comments:

Please discuss the increase of REM sleep by MDL 100907 by providing possible explanations or other references. Several papers indicated the increase in NREM sleep in rats and humans, but there was no change in REM sleep.

There was no schematic diagram of sciatic nerve ligation, planter test, and especially EEG EMG implantation.

Please correct Figure 5 legend and some other spelling.

In Figure 4 there was a visible difference (not significant) at 22:00 between vehicle and amitriptyline group. How did the authors define this? However, the authors defined nerve ligation mice showed light sleep from the beginning and kept sleep pattern flat but if we observe from 13:00 to 22:00 time period, this is not correct, and please explain more precisely about this.

Reviewer #3: The manuscript "Analysis of the effect of tricyclic antidepressant on secondary sleep disturbance induced by chronic pain" by Ito et al., addresses the important connection between chronic pain and sleep. They use a mouse model (sciatic nerve ligation) to induce chronic pain and find a persistent increase in pain sensitivity in these mice after one week. The authors show that this increase in pain sensitivity can be reduced with the treatment of the tricyclic antidepressant Amitriptyline. Furthermore, they show that mice with sciatic nerve ligation spend more time awake and less time in NREM sleep over a 24h period than fellow control mice. Administration of Amitriptyline significantly decreases the amount of time spent in wake and increased NREM sleep amounts, showing that Amitriptyline can increase NREM sleep quantity. Next, Ito and co-workers show that Amitriptyline also improves sleep quality in that the drug increases EEG delta power in ligated mice during the first 3 hours of the light phase. Lastly, to try to elucidate some of the mechanisms through which Amitriptyline might affect sleep and pain, the authors use a selective 5HT2A receptor antagonist (MDL100907) and show that this drug also improves sleep quantity and quality in ligated mice, but fails to decrease pain sensitivity. The authors provide a step by step framework for their experiments that is easy to follow.

Major points:

1) While on first glance well executed, this study does not provide truly new insights into the interrelationship between sleep and pain or into the role of Amitriptyline. Previous studies (e.g. Boyle et al., 2012 Diabetes Care) already show that Amitriptyline is effective at treating pain and improving sleep in humans. The new information here is that Amitriptyline might work though the 5-HTA2 receptor to achieve its effects on sleep and pain. However, the selective 5HT2A receptor antagonist MDL100907 used here to support this claim does not improve pain sensitivity, thereby casting doubts on the involvement of the 5-HT2A involvement.

2) The authors stress the idea that Amitriptyline might increase NREM sleep and reduce pain sensitivity by blocking the 5-HTA2 receptor, but do not discuss that Amitriptyline is a tricyclic anti-depressant and is thought to primarily inhibit the re-uptake of Serotonin and Norepinephrine (thereby increasing the concentrations of these neurotransmitters in the synaptic cleft). This important point needs to be addressed and citations for the antagonistic effect of Amitriptyline on the 5-HTA2 receptor need to be added.

3) The authors mention in the methods that the implantation of the EEG/EMG leads happened 2 days before baseline sleep recordings with a 24h acclimation period to the recording chamber and cables. This time frame seems seriously short considering the severity of the surgery and the discomfort experienced thereafter. Never mind that animals were then connected to EEG/EMG cables to which they only had 24h to acclimate. Another point is the effect of anesthesia (such as Isoflurane) on subsequent sleep/wake architecture soon after the exposure (e.g. Pick et al., 2011 Anesthesiiology; Zhang et al., 2016 Sleep). Considering how susceptible sleep is to environmental influences (including pain and discomfort), it would have been better to first perform the EEG/EMG surgery, acclimate the mice to the cage and cable for 1-2 weeks total (as is otherwise customary in mouse sleep research) and then perform the sciatic nerve ligation and drug applications.

Minor:

Figure 2 – please show individual data points to allow a more informative assessment of the results

Figure 4 – shows “normalized delta power density’ for each mouse and demonstrates a flat beginning for vehicle-treated ligation mice. However, it would also be informative to compare the delta amplitude in more absolute values to see whether nerve ligation as such decreased delta during NREM sleep as compared to the sham surgery. Also, the “sham+amitriptyline” group is missing from this graph.

Please format references to show complete spelling of last names, rather than first names (this happened in several references)

To improve readability of the manuscript , the authors should work with a language coach to address spelling and sentence structure throughout the manuscript

Some important references could be added in the introduction e.g. Alexandre et al., 2017 Nat Med

6. PLOS authors have the option to publish the peer review history of their article (what does this mean?). If published, this will include your full peer review and any attached files.

Reviewer #1: No

Reviewer #2: No

Reviewer #3: No

---

## [Author Response · Author response to Decision Letter 0]

25 Oct 2020

Journal Requirements:

A. We have checked our manuscripts to meet PLOS ONE's style.

2. Thank you for including your ethics statement: "The present study was conducted in accordance with the Guiding Principles for the Care and Use of Laboratory Animals at University of Toyama (Toyama, Japan), as adopted by the Committee on Animal Research of University of Toyama.

Approval number: A2014NED-14

Methods of anesthesia: general anesthesia using isoflurane

Methods for euthanasia of animals: Carbon dioxide and Nerve destruction". 

Please amend your current ethics statement to confirm that your named ethics committee specifically approved this study.

A. We have amended our ethics statement. (p.5 Line 75)

Response to Reviewers 

Reviewer #1: The authors have provided an interesting study that contributes to the growing field of knowledge surrounding the relationship between disrupted sleep and chronic pain. Specifically, the authors sought to examine the role of serotonergic components of the ascending reticular activating system in sleep disruption induced by chronic pain. To accomplish this, a partial sciatic nerve ligation model was selected and a tricyclic antidepressant (amitriptyline) and a selective 5-HT2A antagonist (MDL 100907) were deployed. Using the plantar test and EMG/EEG recording, the authors observed that amitriptyline improved both hyperalgesia symptoms and sleep quantity, whereas MDL100907 only normalized sleep quantity but had no effects on examined hyperalgesia symptoms. From this study, we can conclude that that amitriptyline’s effects on improved sleep quality in rodents during chronic pain is likely due to at least some antagonism of 5-HT2AR, but that its ability to ameliorate hyperalgesia is mediated by another mechanism unrelated to 5-HT2AR antagonism. The authors provide a strong list of limitations to the interpretation of this study. 

However, the authors need to seek out a native English speaker to improve the quality of the writing and grammar, need to explain why several groups appear to have been excluded from analysis (See Results comments), why proper controls were not used throughout (See Results comments), and why only males were used for the study (and please do not say because of the cyclic female sex steroid hormones unless they address the males’ pulsatile testosterone secretion patterns).

Comments:

Introduction:

1. In the introduction, the authors should provide rationale for why they selected amitriptyline as the TCA of choice from the cited study (16). It may also be useful for the reader to know that it is commonly prescribed in the treatment of neuropathic pain. 

A. Thank you for highlighting this point. According to your advice, we added the following statement: “Amitriptyline, a tricyclic antidepressant, is a first-line drug for neuropathic pain, and is commonly prescribed for the treatment of neuropathic pain.” in the introduction section of our manuscript. (p.4 Line 51) 

2. In the introduction, the rationale for using MDR 100907 should be explained. The third paragraph of the discussion may function best if moved to the introduction.

A. Thank you for your advice. We have moved the third paragraph of the discussion to the introduction. (p.4 Line 57)

- Authors should provide justification for using only males in the study.

A. Thank you for the helpful suggestion. As you rightly indicated, the investigation of females is equally important. However, as the mechanisms of chronic pain have been shown to differ between males and females, we need to examine them separately. Although we investigated using only males in this study, we consider that similar studies on females should also be conducted. We mentioned this as a limitation of this study in the discussion section. (p.20 Line 36)

Methods:

- Did the mice receive any analgesic after surgery? If so, then the authors should describe this. 

A. We did not use any analgesics after surgery in this study. 

- The methods describing the evaluation of hyperalgesia are not consistent with the timelines in figures 2 & 3. Specifically, “Immediately after the administration of amitriptyline, selective 5-HT2A receptor antagonist, MDL100907, or vehicle, mice were placed in the acrylic cylinder (height: 15cm, Diameter: 8cm) and habituated for 1 hour before measurement.” These discrepancies should be corrected. 

A. Thank you for pointing out the discrepancy. We have specified the timeline of plantar tests according to the figures. (p.7 Line 95)

- What was the inter-stimulus interval for the repeated measurements of paw withdrawal latency? (E.g., 3-minutes were given between each test).

A. We specified that we had set the test interval of 3 minutes or more. (p.7 Line 102)

- The authors do not discuss ipsilateral or contralateral paw tests. Considering this information is not provided, we must assume that only the ipsilateral paw was tested. This should be clarified by the authors and a justification for not testing the contralateral paw should be provided. 

A. We specified that we tested only the ipsilateral paw in the plantar tests. (p.7 Line 104) The most important aim of the plantar test in this study was to confirm the effects of sciatic nerve ligation surgery and drug administrations on pain in the injured paw. The ipsilateral assessment is sufficient to determine whether the sciatic nerve is successful or if the drug improves the pain. We have determined that the contralateral test is less useful in this study and such less needed painful tests are unethical.

- Though commonly used, the authors should cite the methods of classifying REM vs. NREM via Theta/Delta ratios, (e.g. Grosmark et. Al 2012 Cell).

A. We added a brief description of the methods of classifying REM and non-REM in brief, along with a relevant citation [23] (p.8 Line 125).

Results:

- Figure 2 and figure 5 do not contain proper controls (i.e. a sham PSL group). The authors should provide a strong justification as to why this group was not included and a within-group historic control was used instead. Additionally, the authors should provide justification as to why a saline injection was given to the same group for a within-group historical comparison as opposed to having a full-factorial study design.

A. As you indicated, it is necessary to show the sham data. We have added the data from sham experiments in figure 2 and 5.

Because sciatic nerve ligation causes stressful chronic pain in animals, we are required to use the minimum number of animals due to ethical issues. To overcome this restriction, we administered vehicle and drug to the same group for historical comparison. Following the advice of reviewer 2, we confirmed that the mouse showed the same hyperalgesia on the test days (days 7, 8, and 9). We changed figure 1 to show these data. 

- Authors should delete word “clearly” from results section 1.

A. We have deleted “clearly” from this section.

- Authors should provide rationale for why the sham-drug groups were excluded from figures4 and 7.

A. Thank you pointing out the omission. We have added the data from sham-drugs groups in figures 4 and 7.

- Authors should explain that delta power was increased during the inactive but not active phase. Reword this sentence “By administration of amitriptyline, nerve ligation mice got deep sleep at first and went to light sleep as same as sham mice” in Results section 3.

A. Thank you for your advice. We reworded the sentences, “With administration of amitriptyline, nerve ligation mice got deep sleep at first, in the inactive phase. Although there were no significant differences, amitriptyline tended to reduce the δ wave power in the active phase” in results section 3 (p.13, Line 205)

Discussion:

- Authors should reword the following sentence “According to the time change of normalized power density of Δ wave during non-REM sleep, nerve ligation group with vehicle administration did not get enough had less deep sleep at an early sleep phase during the beginning of the inactive

phase”

A. We have reworded the sentence pointed out and the sentence following it as follows,

“According to the time change of normalized power density of δ wave during non-REM sleep, in the nerve ligation group with vehicle administration, the non-REM sleep is not deep enough in the beginning of the inactive phase. On the other hand, the nerve ligation group with amitriptyline administration showed a time change of normalized power density of δ wave similar to the sham-vehicle group in the inactive phase. These findings suggest that amitriptyline could provide near-normal quality of sleep in chronic pain.” (p.18, Line 281)

Reviewer #2: Antidepressants are reported to have analgesic effects, especially on chronic pain, and tricyclic antidepressants are used to treat chronic pain in combination with serotonin-noradrenaline reuptake inhibitors. Here, Ito and colleagues used the amitriptyline, a tricyclic antidepressant, and MDL 100907, a selective 5-HT2Aantagonist, affect the reduction of chronic pain-induced sleep disturbance.

My major concern regarding this manuscript is about the experimental design. 

The authors confirmed the paw withdrawal latency of thermal hyperalgesia in neuropathic pain model mice in a weekly manner (7 days apart). However, in the subsequent experiments, they recorded the effect either on the next day (day 7, 8) or after of gap of 2 days (day7, 9) of the vehicle administration. 

Authors can follow any of the following three alternatives: 

-Providing evidence that the mouse showed the same paw withdrawal latency within the days; or re-perform the experiment as a crossover manner, and/or collect the data weekly. 

A. Thank you for your pointing out these discrepancies. We re-conducted plantar tests as per the schedule provided by you and changed figure 1 to show the same paw withdrawal latency from day 7 to day 9.

Moreover, the authors have given less than 2 days for recovery after EEG mounting which might not be enough. 

A. In previous studies, which used the same EEG recording devices (Pinnacle technology, USA), recovery time given was 2 days (references 12, 15). This EEG mounting surgery is less invasive and only needs a short anesthesia time < 15 minutes. Therefore, we set the recovery time from surgery as 2 days, which was the same as in previous studies. However, we could not confirm the presence or absence of any effects of stress caused by the surgery and anesthesia.

We added the above text in the discussion section as a limitation. (p.20 Line 316)

The data presented to show the effect of the drug on sleep is also not convincing. Please provide a full frequency of EEG analysis in each vigilance stage of vehicle administration. 

A. Thank you very much for your advice. We have added the results of full frequency of EEG analysis in result section 3 (p.13 Line 198) and 5 (p.16 Line 249) and included new figures 4A and 7A. 

“There were slight differences in peak of EEG power density during non-REM sleep when comparing sham-vehicle and ligation-vehicle in the amitriptyline experiment (Figure 4A), but only a tendency in the MDL experiment (Figure 7A). The reduction of sleep time with nerve injury may increase the absolute value of δ power density slightly, as homeostatic compensation [31]. However, the results were not reproducible in our study.” 

We have added the above text in the discussion section (p.19 Line 302).

Please provide the details on how they calculate the mean time used in each figure.

A. non-REM sleep time, REM sleep time, and awakening time per 24 hours for each animal were calculated by SleepSign. The values were summed for each group and divided by the number of animals to calculate the mean for each group. Finally, we calculated the mean time per hour.

We described this calculation of mean time in methods section 5. (p.8 Line 127) 

Minor Comments:

Please discuss the increase of REM sleep by MDL 100907 by providing possible explanations or other references. Several papers indicated the increase in NREM sleep in rats and humans, but there was no change in REM sleep. 

A. “5-HT2A receptor antagonist slightly increased the time of REM sleep in both the ligation and sham groups in this study. There are few reports examining the effects of 5-HT2A receptor antagonists on REM sleep, and their results are controversial. Jaime M. et al reported that the administration of 5-HT2A antagonist reversed the reduction of REM sleep caused by microinjection of 5-HT2A/2C receptor agonist into the dorsal raphe nucleus in rat [30]. However, Stephen R. et al reported that 5-HT2A antagonist tended to increase REM sleep compared with vehicle administration, but not significantly [27].” This is contrary to what you have mentioned 

We added above contents in discussion. (p.19 Line 295)

There was no schematic diagram of sciatic nerve ligation, planter test, and especially EEG EMG implantation. 

A. Following your advice, we added schematic diagrams of sciatic nerve ligation (Fig 1A), plantar test (Fig 2B), and EEG EMG implantation (Fig 3B).

Please correct Figure 5 legend and some other spelling. 

A. We have corrected them. A professional English language editing service also have checked all. 

In Figure 4 there was a visible difference (not significant) at 22:00 between vehicle and amitriptyline group. How did the authors define this? 

A. As you pointed out, there was a visible difference at 22:00 in the amitriptyline group. We added the following sentences in the second paragraph of the discussion.

“Amitriptyline tended to decrease the delta power at 22:00, though not significantly, in both sham and ligation groups. Therefore, amitriptyline may induce shallow sleep in the active phase. However, since there is less sleep in the active phase, the data dispersion is large, and further research on this aspect is needed .” (p. 18 Line 286)

However, the authors defined nerve ligation mice showed light sleep from the beginning and kept sleep pattern flat but if we observe from 13:00 to 22:00 time period, this is not correct, and please explain more precisely about this. 

A. Thank you for pointing out this discrepancy. As you mentioned, the sentence did not convey the observations correctly. We reworded the sentence as below. (p.13 Line 203)

“Sleep of nerve ligation mice was shallow in the beginning and deepened gradually until 16:00. However, there was no significant difference in the dark phase.”

Reviewer #3: The manuscript "Analysis of the effect of tricyclic antidepressant on secondary sleep disturbance induced by chronic pain" by Ito et al., addresses the important connection between chronic pain and sleep. They use a mouse model (sciatic nerve ligation) to induce chronic pain and find a persistent increase in pain sensitivity in these mice after one week. The authors show that this increase in pain sensitivity can be reduced with the treatment of the tricyclic antidepressant Amitriptyline. Furthermore, they show that mice with sciatic nerve ligation spend more time awake and less time in NREM sleep over a 24h period than fellow control mice. Administration of Amitriptyline significantly decreases the amount of time spent in wake and increased NREM sleep amounts, showing that Amitriptyline can increase NREM sleep quantity. Next, Ito and co-workers show that Amitriptyline also improves sleep quality in that the drug increases EEG delta power in ligated mice during the first 3 hours of the light phase. Lastly, to try to elucidate some of them echanisms through which Amitriptyline might affect sleep and pain, the authors use a selective 5HT2A receptor antagonist (MDL100907) and show that this drug also improves sleep quantity and quality in ligated mice, but fails to decrease pain sensitivity. The authors provide a step by step framework for their experiments that is easy to follow.

Major points:

1) While on first glance well executed, this study does not provide truly new insights into the interrelationship between sleep and pain or into the role of Amitriptyline. Previous studies (e.g. Boyle et al., 2012 Diabetes Care) already show that Amitriptyline is effective at treating pain and improving sleep in humans. The new information here is that Amitriptyline might work though the 5-HTA2 receptor to achieve its effects on sleep and pain. However, the selective 5HT2A receptor antagonist MDL100907 used here to support this claim does not improve pain sensitivity, thereby casting doubts on the involvement of the 5-HT2A involvement. 

A. I really appreciate your pointing out this important problem.

As you mentioned, amitriptyline is the first-line treatment for chronic neuropathic pain and could improve sleep disturbance induced by chronic pain at the same time. However, it is not known whether this sleep-improving effect is an indirect effect due to pain improvement or a direct pharmacological effect of amitriptyline (p.4 Line 51). We hypothesized that antagonism of excitatory 5-HT2A receptors of amitriptyline would improve pain-related sleep disturbance, because the activation of ascending serotoninergic neurons could be partly associated with the mechanism of sleep disorders caused by chronic pain from our previous study (23) (p.5 Line 65). In the results of this study, antagonism of 5-HT2A receptors was shown to improve sleep disturbance in the chronic pain model “without improving pain”; therefore, it was speculated that amitriptyline could have a direct sleep-improving effect in addition to improving pain. (p.19 Line 294)

 Based on your suggestion, we have carefully modified the introduction and discussion.

2) The authors stress the idea that Amitriptyline might increase NREM sleep and reduce pain sensitivity by blocking the 5-HTA2 receptor, but do not discuss that Amitriptyline is a tricyclic anti-depressant and is thought to primarily inhibit the re-uptake of Serotonin and Norepinephrine (thereby increasing the concentrations of these neurotransmitters in the synaptic cleft). This important point needs to be addressed and citations for the antagonistic effect of Amitriptyline on the 5-HTA2 receptor need to be added.

　You have raised an important concern. First, we would like to say that amitriptyline might increase NREM sleep by blocking the 5-HT2A receptor, but refrain from saying that this mechanism is associated with improving pain. We have modified the manuscript to avoid any misinterpretation of our meaning. 

 Second, as per your suggestion on discussing the main pharmacological action of amitriptyline, i.e. inhibition of re-uptake of serotonin and norepinephrine, we have added the following text to the discussion. “Since amitriptyline inhibits the re-uptake of serotonin and norepinephrine, it increases the concentrations of these neurotransmitters in the synaptic cleft. However, amitriptyline antagonized excitatory 5-HT2A receptors coupled with Gq protein induce spontaneous excitatory synaptic transmission. On the other hand, amitriptyline has little effect on the inhibitory 5-HT1A receptors coupled with Gi protein. Therefore, we believe that amitriptyline may suppress the neuronal activity as an overall effect.” (p.17 Line 270) 

 Lastly, following your advice, we have added the citations, of the studies of receptor binding assay that show amitriptyline antagonize 5-HT2A receptor but not 5-HT1A [21, 22]. (p.18 Line 274)

3) The authors mention in the methods that the implantation of the EEG/EMG leads happened 2 days before baseline sleep recordings with a 24h acclimation period to the recording chamber and cables. This time frame seems seriously short considering the severity of the surgery and the discomfort experienced thereafter. Never mind that animals were then connected to EEG/EMG cables to which they only had 24h toacclimate. Another point is the effect of anesthesia (such as Isoflurane) on subsequent sleep/wake architecture soon after the exposure (e.g. Pick et al., 2011 Anesthesiiology; Zhang et al., 2016 Sleep). Considering how susceptible sleep is to environmental influences (including pain and discomfort), it would have been better to first perform the EEG/EMG surgery, acclimate the mice to the cage and cable for 1-2 weeks total (as is otherwise customary in mouse sleep research) and then perform the sciatic nerve ligation and drug applications.

A. Thank you for pointing out the important limitation.

In previous studies which used the same EEG recording devices (Pinnacle technology, USA), recovery time given was 2 days [13, 23]. This EEG mounting surgery is less invasive and only needs a short anesthesia time < 15 minutes. Therefore, we set the recovery time from surgery as 2 days, which is the same as in the previous studies. However, we could not confirm the presence or absence of any effects of stress caused by the surgery and anesthesia. 

We have added the above text in the discussion as a limitation. (p.20 Line 316)

Minor: 

Figure 2 – please show individual data points to allow a more informative assessment of the results

A. We have changed figures 2 and 5 to show the individual data as per your advice.

Figure 4 – shows “normalized delta power density’ for each mouse and demonstrates a flat beginning for vehicle-treated ligation mice. However, it would also be informative to compare the delta amplitude in more absolute values to see whether nerve ligation as such decreased delta during NREM sleep as compared to the sham surgery. 

A. Thank you for your helpful comment. We have analyzed the full frequency of EEG and included new Figures 4A and 7A. However, PSNL-vehicle groups showed a slightly higher peak of EEG power density than sham during non-REM in the amitriptyline-experiment (Figure 4A), but only a tendency in the MDL-experiment (Figure 7A). The reduction of sleep time with nerve injury may increase the absolute value of delta power density slightly, as a homeostatic compensation [31]. However, the results were not reproducible in our study.” 

We have added the above text to the discussion (p.19 Line 302).

Also, the “sham+ amitriptyline” group is missing from this graph.

A. We have included the sham+ amitriptyline group in figures 4 and 7. 

Please format references to show complete spelling of last names, rather than first names (this happened in several references) 

A. Thank you for your pointing out it. We have formatted references carefully.

To improve readability of the manuscript, the authors should work with a language coach to address spelling and sentence structure throughout the manuscript. 

A. Thank you for your suggestion. We have opted for a professional English language editing service to check our manuscript and ensure readability.

Some important references could be added in the introduction e.g. Alexandre et al., 2017 Nat Med

A. We have included the paper you recommended in the references [7].

---

## [Decision Letter · Decision Letter 1]

19 Nov 2020

Analysis of the effects of a tricyclic antidepressant on secondary sleep disturbance induced by chronic pain in a preclinical model

PONE-D-20-20832R1

Dear Dr. 

We’re pleased to inform you that your manuscript has been judged scientifically suitable for publication and will be formally accepted for publication once it meets all outstanding technical requirements.

Kind regards,

Rosanna Di Paola, MD

Academic Editor

PLOS ONE

Additional Editor Comments (optional):

Reviewers' comments:

Reviewer's Responses to Questions

**Comments to the Author**

1. If the authors have adequately addressed your comments raised in a previous round of review and you feel that this manuscript is now acceptable for publication, you may indicate that here to bypass the “Comments to the Author” section, enter your conflict of interest statement in the “Confidential to Editor” section, and submit your "Accept" recommendation.

Reviewer #1: All comments have been addressed

Reviewer #3: All comments have been addressed

2. Is the manuscript technically sound, and do the data support the conclusions?

Reviewer #1: Yes

Reviewer #3: Yes

3. Has the statistical analysis been performed appropriately and rigorously? 

Reviewer #1: Yes

Reviewer #3: Yes

4. Have the authors made all data underlying the findings in their manuscript fully available?

Reviewer #1: Yes

Reviewer #3: Yes

5. Is the manuscript presented in an intelligible fashion and written in standard English?

Reviewer #1: Yes

Reviewer #3: Yes

6. Review Comments to the Author

Reviewer #1: The authors have addressed my previous concerns. They said they used a professional English grammar consultant, but there are still some issues.

Reviewer #3: All comments were addressed and the manuscript is greatly improved - ready for publication.

In future studies though, please consider allowing more time for mice to acclimate to EEG/EMG cables. In my experience even after 4 days, mice still do not show the same amplitude of delta NREM sleep that they do after 2 weeks. So perhaps implant the EEG/EEG first, acclimate, and then perform other manipulations.

7. PLOS authors have the option to publish the peer review history of their article (what does this mean?). If published, this will include your full peer review and any attached files.

Reviewer #1: No

Reviewer #3: No

---

## [Editor Report · Acceptance letter]

23 Nov 2020

PONE-D-20-20832R1 

Analysis of the effects of a tricyclic antidepressant on secondary sleep disturbance induced by chronic pain in a preclinical model 

Dear Dr. Ito:

I'm pleased to inform you that your manuscript has been deemed suitable for publication in PLOS ONE. Congratulations! Your manuscript is now with our production department. 

Kind regards, 

on behalf of

Dr. Rosanna Di Paola 

Academic Editor

PLOS ONE